


# Numerical modelling of the tides in the Caspian Sea

Igor Medvedev[1,2], Evgueni Kulikov[1], Isaac Fine[3]

[1]Shirhov Institute of Oceanology, Russian Academy of Sciences, Moscow, Russia

[2]Fedorov Institute of Applied Geophysics, Moscow, Russia

[3]Institute of Ocean Sciences, Sidney, B.C., Canada

*Correspondence to*: Igor Medvedev (patamates@gmail.com)

**Abstract.** The Caspian Sea is the largest enclosed basin on the Earth and a unique object for analysis of tidal dynamics. The Caspian Sea has independent tides only, which are generated directly by tide-forming forces. Using the Princeton Ocean Model (POM), we examine the spatial

and temporal features of tidal dynamics in the Caspian Sea in detail. We present tidal charts for amplitudes and phase lags of the major tidal harmonics, form factor, tidal range and velocity of tidal currents. Semidiurnal tides in the Caspian Sea are determined by a Taylor amphidromic system with counterclockwise rotation. The largest $M_2$ amplitude is 6 cm and is located in the Turkmen Bay. The Absheron Peninsula splits this system into two separate amphidromies with

counterclockwise rotation to the north and to the south of it. The maximum $K_1$ amplitudes (up to 0.7–0.8 cm) are located in: 1) the southeastern part of the Caspian Sea, 2) the Türkmenbaşy Gulf, 3) the Mangyshlak Bay, and 4) the Kizlyar Bay. The semidiurnal tides prevail over diurnal tides in the Caspian Sea. The maximum tidal range has been observed in the Turkmen Bay, up to 21 cm. The highest velocity of the total tidal currents is observed in the straits to the north and south

of Ogurja Ada, up to 22 cm/s and 19 cm/s, respectively. Were made numerical experiments with tidal simulation using different mean sea level MSL of the Caspian Sea (from -25 m to -30 m). Numerical experiments indicate that the spatial features of tides are strongly sensitive to the MSL changes.

## 1 Introduction

Tides, one of the major types of ocean water motion, are formed under the influence of tide-generating forces of the Moon and the Sun and the rotation of the Earth. Tides can be represented as the sum of two types of oscillations: (1) the co-oscillating tide caused by the tidal influx from an adjacent basin, and (2) the independent tide, which is generated directly by the tide-generating forces (Defant, 1961). Co-oscillating tides dominate in marginal seas, generated by

tidal waves penetrating from the adjoining ocean or seas. In isolated inland seas (e.g., the Black Sea and the Baltic Sea), independent tides strongly prevail as tidal waves from adjacent basins cannot significantly penetrate the sea (Medvedev et al., 2013; Medvedev et al., 2016; Medvedev,





2018). The Caspian Sea is a unique object for analysis of independent tide formation as it is the largest enclosed basin on Earth.

Tides in the Caspian Sea have been studied for a long time, though not on a regular basis. Malinovsky (1926) showed that the semidiurnal tides dominate in the Caspian Sea and the spring
tidal range is 7.7 cm based on analysis of 30-day hourly records at three tide gauges. German (1970) performed spectral analysis of three-month observational series at eight tide gauges and distinguished the diurnal and semidiurnal constituents with different geneses: semidiurnal tides have a gravitational origin while diurnal tides are formed by sea-breezes. Kosarev and Tsyganov (1972) and Spidchenko (1973) estimated the amplitudes and phase lags at different sites. Having
analyzed annual series of hourly observations at six tide gauges, Levyant et al. (1994) hypothesized that a semidiurnal tidal wave represents a counterclockwise amphidromic system (like a Kelvin wave) with a center in the Apsheron Threshold's area.

Medvedev et al. (2017) estimated the amplitudes and phase lags of major tidal constituents for different sea parts based on long-term hourly data analysis from 12 tide gauges. Maximum
tidal range of 21 cm was found at the Aladga (eastern part of the South Caspian). Medvedev et al. (2017) also performed high-resolution spectral analysis and determined that the diurnal sea level oscillations in the Middle Caspian have a gravitational origin, while those in the South Caspian are mainly caused by radiational effects: the amplitude of diurnal radiational constituent S1 is much higher than those of gravitational constituents $O_1$, $P_1$, and $K_1$. In the North Caspian, there
are no gravitational tides and only weak radiational tides are observed. A semidiurnal tide is predominant in the Middle Caspian and in the South Caspian.

An analysis of tide gauge data allows for the examination specific tidal features at different sites, but not for the estimation of the spatial structure of tides in deep-water areas of the Caspian Sea. Therefore, in order to capture these spatial structures we adapted the numerical Princeton
Ocean Model (POM) to the Caspian Sea in (Medvedev et al., 2019). The developed POM reproduces the meteorological sea level variability with periods ranging from several hours to a month and tides in the Caspian Sea, which we can use to describe in detail the spatial and temporal peculiarities of tidal dynamics of the whole the Caspian Sea.





**2 Data and methods**

In this study we used 2D version of the Princeton Ocean Model (POM) (Mellor, 2004). The forcing term in two-dimensional shallow water equations was specified through the gradients of tidal potential over the Caspian Sea:

$$\bar{F}_T = -(1 + k - h)\nabla\bar{\Omega}, \tag{1}$$

where $k$ and $h$ are the Love numbers and $\bar{\Omega}$ is the tidal potential. Love numbers $k$ and $h$ relate the body Earth tide (and associated perturbations) to the potential. We used frequency-dependent values of $h$ and $k$ calculated by Wahr (1981) (Table 1). The tidal potential was calculated for spherical harmonics via formulas provided by Munk and Cartwright (1966) and included all the main tidal components ($> 80$), including major diurnal, semi-diurnal, shallow water and long-period constituents. Additionally, our numerical model includes the ocean tidal loading potential obtained from FES2014 (Finite Element Solution tidal model) produced by NOVELTIS, LEGOS and CLS Space Oceanography Division and distributed by AVISO, with support from CNES (http://www.aviso.altimetry.fr/).

**Table 1. Love numbers and the elasticity factors for major tidal constituents (Wahr, 1981; Kantha and Clayson, 2000).**

| Constituent | Frequency (cpd) | $h$ | $k$ |
|---|---|---|---|
| long-period | | 0.606 | 0.299 |
| $Q_1$ | 0.8932 | 0.604 | 0.298 |
| $O_1$ | 0.9295 | 0.603 | 0.298 |
| $P_1$ | 0.9973 | 0.581 | 0.287 |
| $K_1$ | 1.0027 | 0.520 | 0.256 |
| $J_1$ | 1.0390 | 0.611 | 0.302 |
| semidiurnal | | 0.609 | 0.302 |
| shallow | | 0.609 | 0.302 |

Energy dissipation of generated flows is caused by vertical turbulent viscosity. Friction force in the momentum equations is determined by the speed of the bottom flow and the friction coefficient:

$$\left(\tau_{bx}, \tau_{by}\right) = (C_b u_b |\bar{\mathbf{u}}_b|, C_b v_b |\bar{\mathbf{u}}_b|), \tag{2}$$

where $\bar{\mathbf{u}}_b = (u_b, v_b)$ is flow velocity above the bottom boundary layer (which is assumed to be equal to the barotropic velocity $\bar{\mathbf{u}}_b$ for the 2D model), $C_b$ is the bottom friction coefficient which has the following form:

$$C_b = \max\left[\frac{\kappa^2}{(\ln\{0.5H/z_0\})}, 0.0025\right], \tag{3}$$



where $\kappa = 0.4$ is the von Kármán constant, $z_0$ is the bed roughness length. A minimum value for the bottom friction coefficient, $C_b = 0.0025$, was applied in order to avoid having the bottom drag effect vanish when the water depth is very large.

Numerical simulations were performed on a grid of 507 by 659 nodes with a constant step of 1' in latitude and longitude, created from GEBCO bathymetry data of the Caspian Sea with a resolution of 30 arcseconds. In section 3.1, for numerical modelling the mean sea level (MSL) of the Caspian was set at -28 m of the Baltic Height System (BHS, relative to the zero of the Kronstadt gauge). In the numerical experiments in section 3.2, the MSL of the Caspian varied from -25 to -30 m of the BHS. Boundary conditions for the tidal model are zero flow normal to the coast (at a 2 m depth contour).

In (Medvedev et al., 2019), the model was validated by hourly sea level observations at eight tidal gauges in the Caspian Sea (Fig. 1). In (Medvedev et al., 2019), several experiments with different values of the bed roughness length were performed. The best tide reproduction accuracy at the eight sites was obtained at $z_0 = 0.01$ m, which is used here to determine the bottom friction coefficient $C_b$. All presented results of the tidal analysis are given for Greenwich Mean Time (GMT).


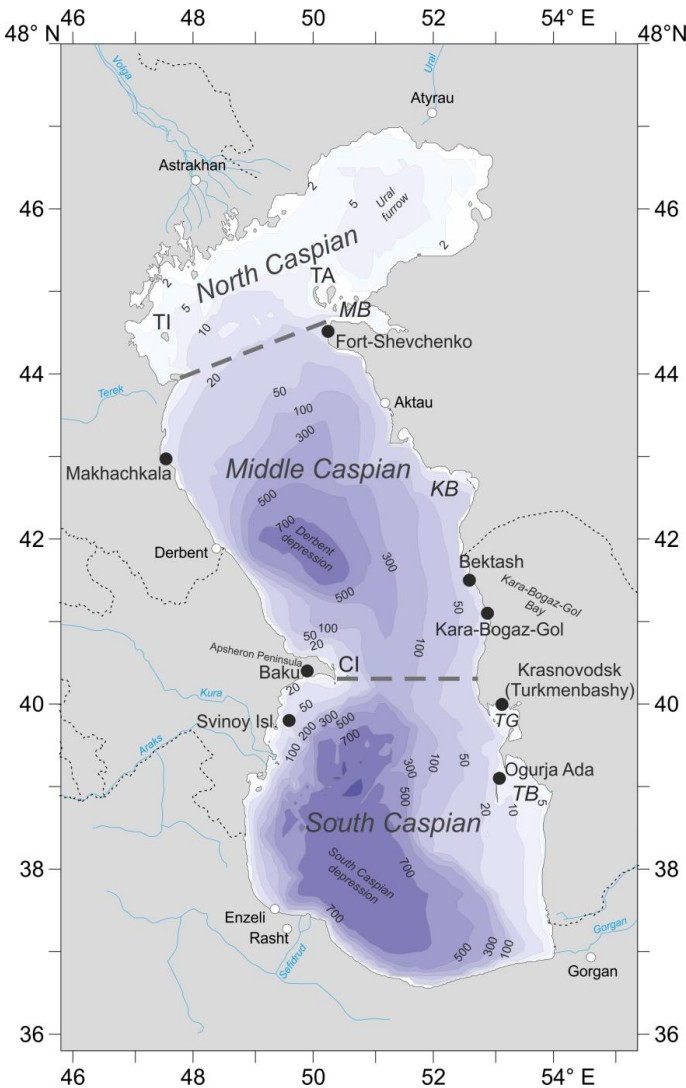

**Figure 1. The bathymetry of the Caspian Sea. Black points are tide gauges used for validation of the numerical model. Other designations: TI is the Tyuleny Island, TA is the Tyuleniy Archipelago, MB is the Mangyshlak Bay, KB is the Kazakh Bay, CI is Chilov (Zhiloy) Island,**
5 **TG is Türkmenbaşy Gulf, TB is Turkmen Bay.**

## 3 Results

### 3.1 Numerical modelling of tides

Numerical model with an MSL of -28 m of the BHS was used in order to reproduce tides

in 1978. Amplitudes and phase lags of major tidal constituents were calculated using classical





harmonic analysis (Pugh and Woodworth, 2014). In this section, we consider the spatial pattern of diurnal and semi-diurnal tides taking major constituents $K_1$ and $M_2$ as an example.

Diurnal tidal pattern includes a complicated amphidromic system in the Middle Caspian (Fig. 2a). The Absheron Peninsula splits this system into two separate amphidromies to the north and south of it. Both amphidromic systems have counterclockwise rotation. Near the Absheron Peninsula, the $K_1$ amplitude is less than 0.15 cm. The maximum $K_1$ amplitudes (up to 0.7–0.8 cm) are located in: 1) the southeastern part of the Caspian Sea, 2) the Türkmenbaşy Gulf, 3) the Mangyshlak Bay, and 4) the Kizlyar Bay. Another amphidromy, with counterclockwise rotation, is formed in the North Caspian. In (Medvedev et al., 2019) we showed that results of numerical modelling in the Northern Caspian are not really trustworthy. For the reason of very shallow depths of this area with about 20% of the North Caspian having depths less than 1 m (Baydin and Kosarev, 1986). Other diurnal tidal constituents have similar spatial distribution to $K_1$. The amplitudes of these constituents are up to 0.5 cm for $O_1$, up to 0.25 cm for $P_1$. The amplitude of the other diurnal tidal constituents in the Caspian Sea does not exceed 0.1 cm.

Semidiurnal tides in the Caspian Sea are determined by a Taylor amphidromic system with counterclockwise rotation. This system is the result of the superposition of two Kelvin waves propagating in opposite directions (Fig. 2b). The amphidromic point of this system is located 80 km east of the Absheron Peninsula. The minimum $M_2$ amplitudes are located in: 1) east of the Absheron Peninsula, 2) western and 3) eastern parts of the North Caspian. The areas with the maximum $M_2$ amplitudes are observed in 1) western part of the South Caspian, up to 2.4 cm; 2) the Kazakh Bay, up to 3.2 cm; 3) the Mangyshlak Bay, up to 3.2 cm; 4) the Türkmenbaşy Gulf, 3.9 cm. The largest $M_2$ amplitude is 6 cm and is located in the Turkmen Bay. Other semidiurnal tidal constituents have a similar spatial distribution to $M_2$. The $S_2$ amplitude in the Turkmen Bay is 2.6 cm, $N_2$ is 1.1 cm, and $K_2$ is 0.7 cm. The amplitudes and phase lags of major tidal constituents at main cities in the Caspian Sea are presented in Table 2.



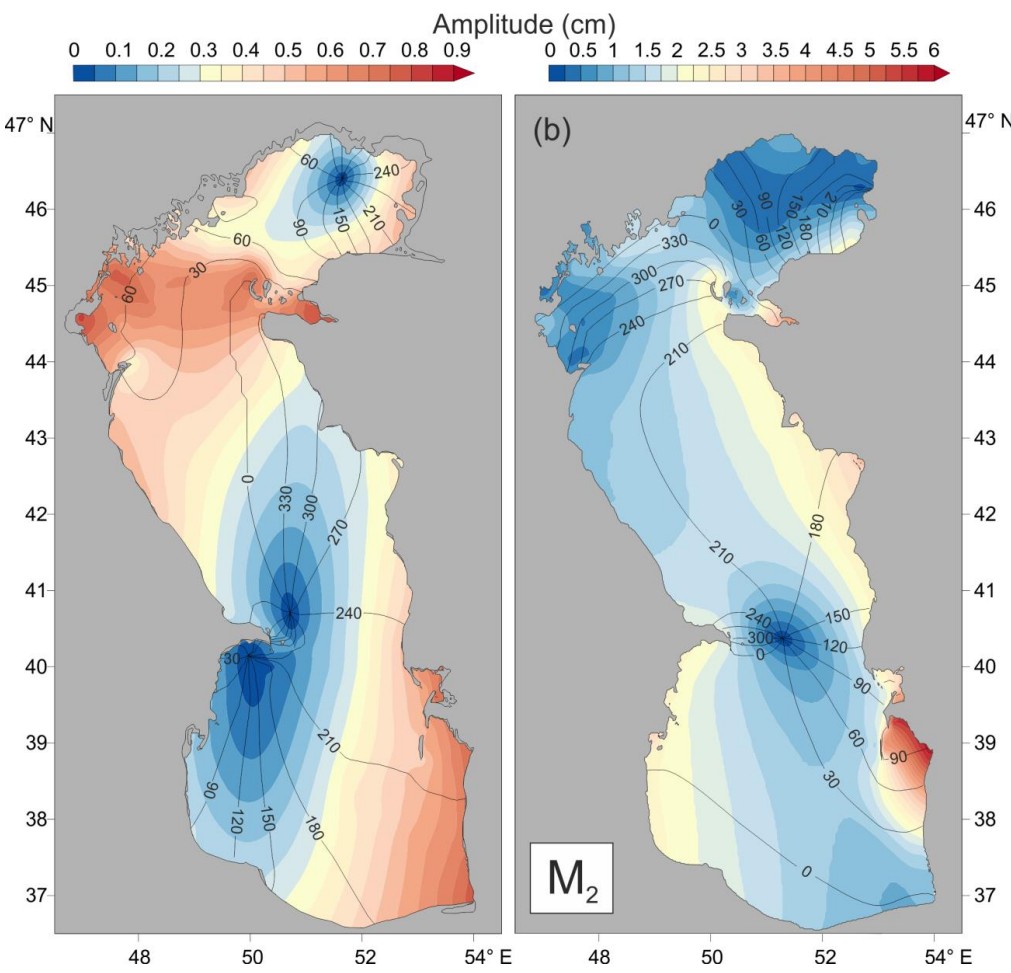

**Figure 2. Tidal maps of co-amplitudes (cm) (shaded) and co-phases (degrees, GMT) (solid lines) for (a) K₁ and (b) M₂.**



**Table 2. Amplitudes (*H*) and Greenwich phase lags (*G*) of major tidal constituents, the form factor (*F*), and maximum tidal range (*R*) at main cities in the Caspian Sea.**

| Station | Country | M₂ | | S₂ | | K₁ | | O₁ | | *F* | *R*, cm |
|---|---|---|---|---|---|---|---|---|---|---|---|
| | | *H*, cm | *G*, ° | *H*, cm | *G*, ° | *H*, cm | *G*, ° | *H*, cm | *G*, ° | | |
| Bandar-e Anzali | Iran | 2.12 | 354 | 0.92 | 354 | 0.24 | 112 | 0.18 | 102 | 0.14 | 7.6 |
| Rudsar | Iran | 1.83 | 353 | 0.77 | 353 | 0.26 | 149 | 0.18 | 139 | 0.17 | 6.6 |
| Tonekabon | Iran | 1.64 | 352 | 0.68 | 352 | 0.33 | 160 | 0.22 | 153 | 0.23 | 6.0 |
| Chalus | Iran | 1.43 | 351 | 0.59 | 351 | 0.41 | 171 | 0.27 | 166 | 0.33 | 5.4 |
| Babolsar | Iran | 1.07 | 357 | 0.41 | 359 | 0.56 | 187 | 0.37 | 184 | 0.62 | 4.7 |
| Bandar Torkaman | Iran | 1.10 | 360 | 0.42 | 10 | 0.78 | 197 | 0.53 | 194 | 0.86 | 5.7 |
| Hazar | Turkmenistan | 1.90 | 88 | 0.74 | 98 | 0.54 | 215 | 0.34 | 213 | 0.33 | 7.0 |
| Türkmenbaşy | Turkmenistan | 2.27 | 129 | 0.85 | 140 | 0.60 | 234 | 0.38 | 230 | 0.31 | 8.2 |
| Garabogaz | Turkmenistan | 2.43 | 167 | 0.99 | 169 | 0.40 | 253 | 0.27 | 248 | 0.19 | 8.6 |
| Aktau | Kazakhstan | 2.30 | 186 | 0.89 | 187 | 0.29 | 303 | 0.18 | 295 | 0.15 | 7.9 |
| Fort Shevchenko | Kazakhstan | 2.47 | 210 | 0.92 | 210 | 0.56 | 326 | 0.30 | 317 | 0.25 | 8.9 |
| Lagan | Russia | 1.16 | 48 | 0.40 | 67 | 0.77 | 75 | 0.44 | 69 | 0.78 | 5.8 |
| Makhachkala | Russia | 1.22 | 228 | 0.48 | 244 | 0.53 | 25 | 0.36 | 23 | 0.52 | 4.9 |
| Kaspiysk | Russia | 1.22 | 229 | 0.48 | 244 | 0.52 | 24 | 0.36 | 23 | 0.51 | 4.9 |
| Izberbash | Russia | 1.23 | 227 | 0.48 | 241 | 0.51 | 23 | 0.35 | 23 | 0.50 | 4.9 |
| Derbent | Russia | 1.27 | 223 | 0.49 | 235 | 0.48 | 25 | 0.33 | 26 | 0.46 | 5.0 |
| Sumqayit | Azerbaijan | 1.80 | 231 | 0.74 | 239 | 0.25 | 28 | 0.17 | 28 | 0.17 | 6.3 |
| Baku | Azerbaijan | 2.18 | 6 | 0.96 | 8 | 0.05 | 358 | 0.03 | 356 | 0.02 | 7.6 |
| Gobustan | Azerbaijan | 2.32 | 7 | 1.03 | 8 | 0.10 | 30 | 0.07 | 18 | 0.05 | 8.2 |
| Neftçala | Azerbaijan | 2.24 | 2 | 1.00 | 3 | 0.13 | 71 | 0.12 | 68 | 0.08 | 8.0 |
| Lankaran | Azerbaijan | 2.40 | 1 | 1.06 | 2 | 0.21 | 71 | 0.18 | 67 | 0.11 | 8.5 |
| Astara | Azerbaijan | 2.29 | 359 | 1.00 | 359 | 0.21 | 79 | 0.18 | 74 | 0.12 | 8.2 |

5  **3.2 Form factor and tidal range**

The results of our analysis indicate that semidiurnal tides prevail over diurnal tides in the Caspian Sea. We estimated the form factor determined by the ratio of major diurnal and semidiurnal constituents (Pugh and Woodworth, 2014):

$$F = \frac{H_{K_1} + H_{O_1}}{H_{M_2} + H_{S_2}} . \tag{4}$$

10  Tides have a semidiurnal form in the eastern part of the Middle Caspian (*F* < 0.25), in the western part of the South Caspian (*F* < 0.25), and in the Turkmen Bay (*F* ~ 0.14, Fig. 3a). In general, mixed mainly semidiurnal tide (0.25 < *F* < 1.5) is observed in other areas of the Caspian Sea. Only in


western and eastern parts of the North Caspian and at the semidiurnal amphidromic point (80 km east of the Absheron Peninsula) the tide has a mixed mainly diurnal form ($F > 1.5$).

Based on the results of numerical modelling of diurnal, semidiurnal and shallow tidal constituents at each grid node the 18.6-year tidal time series have been predicted. The tidal range was calculated as the maximum range of tidal sea level oscillations during one lunar day (~25 hours). The tidal co-range picture has a similar pattern with the $M_2$ amplitude distribution (Fig. 3b). The maximum tidal ranges have been observed in 1) the Kazakh Bay, up to 12 cm; 2) the Mangyshlak Bay, up to 12 cm; 3) the Türkmenbaşy Gulf, 13 cm; 4) the Turkmen Bay, up to 21 cm. The form factor and tidal range at main cities in the Caspian Sea are presented in Table 2.

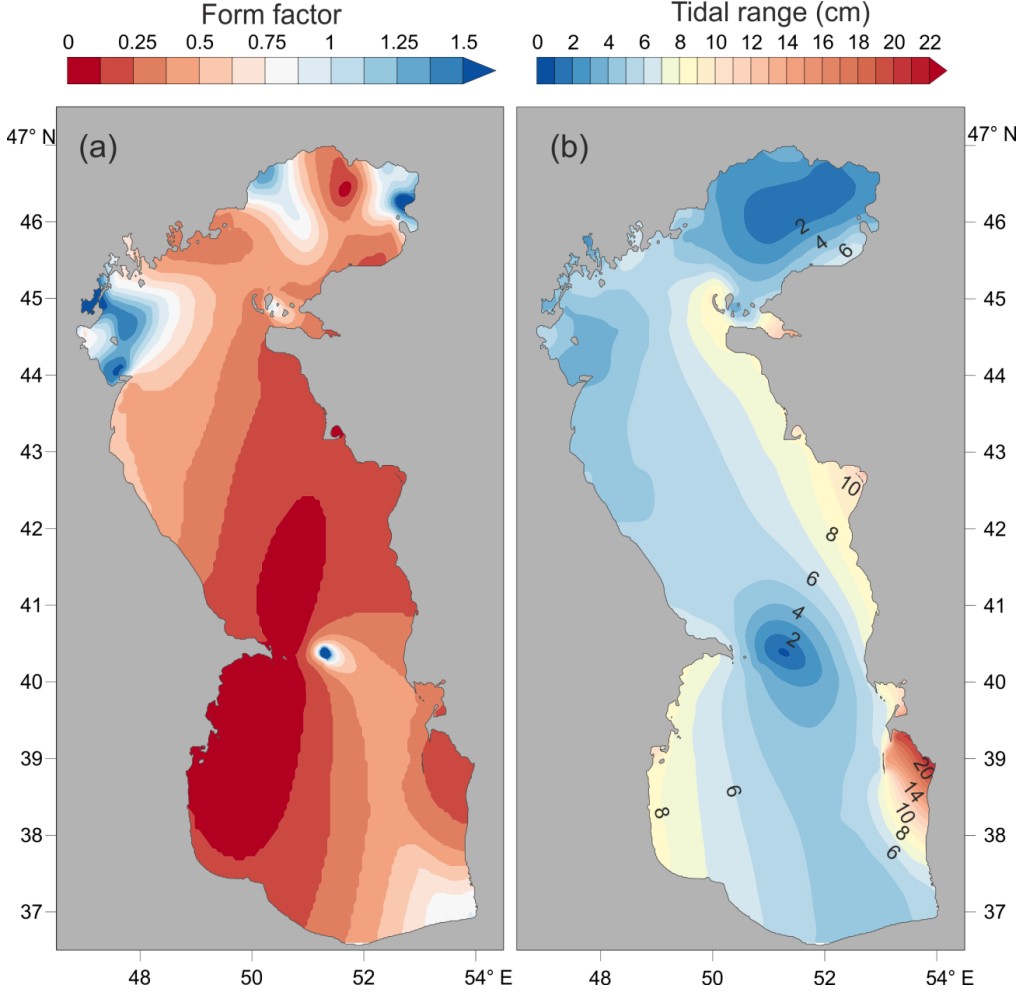

**Figure 3. (a) Form factor and (b) the maximal tidal range in the Caspian Sea.**



### 3.3 Tidal currents

Tidal dynamics are characterized not only by sea level oscillations but also by periodic currents. Spatial structure of the major semi-axis (amplitude) of tidal currents differs from the pattern of the tidal sea level amplitude distribution. Areas of largest $M_2$ currents velocity (major
semi-axis) have been observed (Fig. 4): 1) in the Mangyshlak Bay near the Tyuleniy Archipelago, up to 6.5 cm/s; 2) Absheron Strait which separates the Absheron Peninsula from the Chilov Island, up to 7.5 cm/s; 3) in the straits to the north and south of Ogurja Ada (the Ogurchinsky Island), up to 12.5 and 11.7 cm/s, respectively. The $M_2$ ellipse parameters (major and minor semi-axes, angle of inclination, phase lag) change depending on topographic features of the water area. At the
highest velocity, the rotation of the ellipse occurs in a clockwise direction. In straits and in shallow waters (for example, in the Turkmen Bay), minor semi-axis approaches zero and tidal currents are nearly rectilinear. The spatial pattern of $S_2$ tidal currents in the Caspian Sea has the same structure as $M_2$: the amplification areas and the ellipse parameters remain; only the $S_2$ major semi-axis is 2 times weaker than the $M_2$ major semi-axis. Since $M_2$ and $S_2$ have the largest current velocities in
the Caspian Sea, the spatial pattern of the maximum total tidal currents, calculated from time series computed for 18.6 years, repeats pattern of $M_2$ again. The maximum total tidal current velocity in the Caspian Sea exceeds the $M_2$ velocity on average by 1.8 times. The highest velocity of the total tidal currents is observed mainly in the following straits: 1) the Mangyshlak Bay near the Tyuleniy Archipelago, up to 11.5 cm/s; 2)  Absheron Strait which separates the Absheron Peninsula from
the Chilov Island, up to 13 cm/s; 3) in the straits to the north and south of Ogurja Ada, up to 22 cm/s and 19 cm/s, respectively.


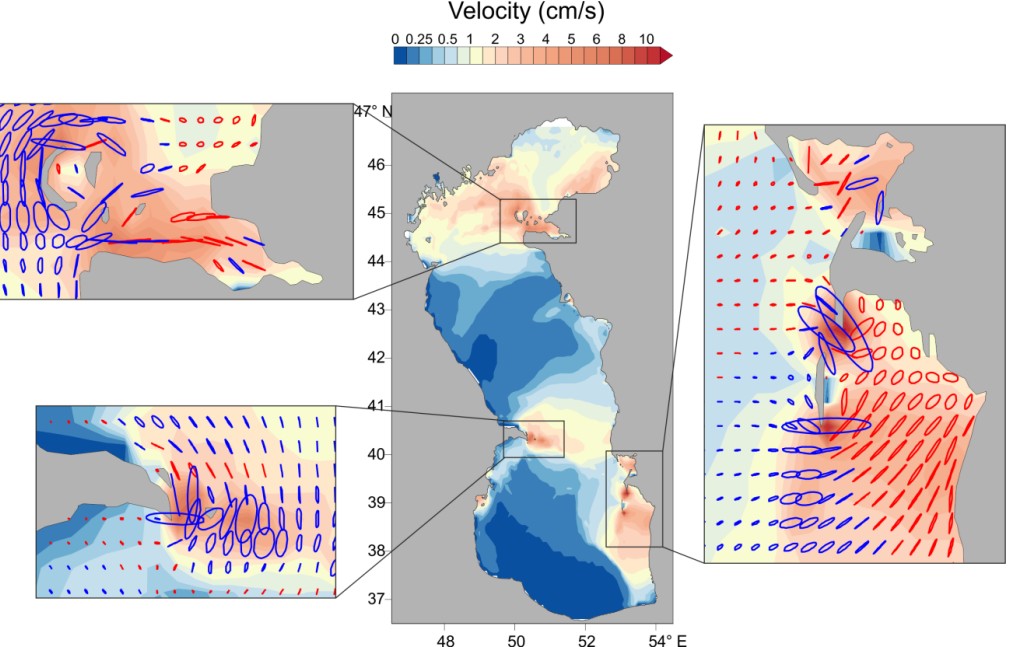

**Figure 4. Major semi-axis magnitudes (cm/s) for M₂ tidal current. Blue ellipses are clockwise and red ellipses are counterclockwise.**

### 3.4 Numerical experiments with different MSL

Interannual MSL variability is one of the main features of the hydrological regime of the Caspian Sea (Bolgov et al., 2007). MSL variations lead to changes in the area and volume of the sea and result in changes in the frequency-selective properties of both the entire Caspian Sea and its individual parts (Fig. 5). The mean depth of the North Caspian is about 5–6 m and 20% of this area has a depth less than 1 m. As a result, the MSL changes of the Caspian Sea by 2–3 m (for

example, from 1974 to 1994) lead to major changes in the sea dynamics of both North Caspian and coastal waters of Middle and South Caspian. Due to long-term MSL changes, spatial characteristics of natural oscillations of the basin (seiches) and the tidal pattern should also change.

In the present study, we made numerical experiments with tidal simulation using different MSL of the Caspian Sea: from -25 m to -30 m of the BHS. It is the natural range of MSL changes

of the Caspian Sea under climatic conditions typical for the Sub-Atlantic climatic interval of the Holocene epoch ("risk zone", Bolgov et al., 2007). Results of these experiments allowed us to estimate the changes in tidal patterns of the Caspian Sea throughout the 19th–20th centuries. Numerical results showed that MSL changes in the course of those centuries led to a significant restructuring of the spatial structure of natural sea level oscillations of the whole sea and its

individual parts (Middle and South Caspian).

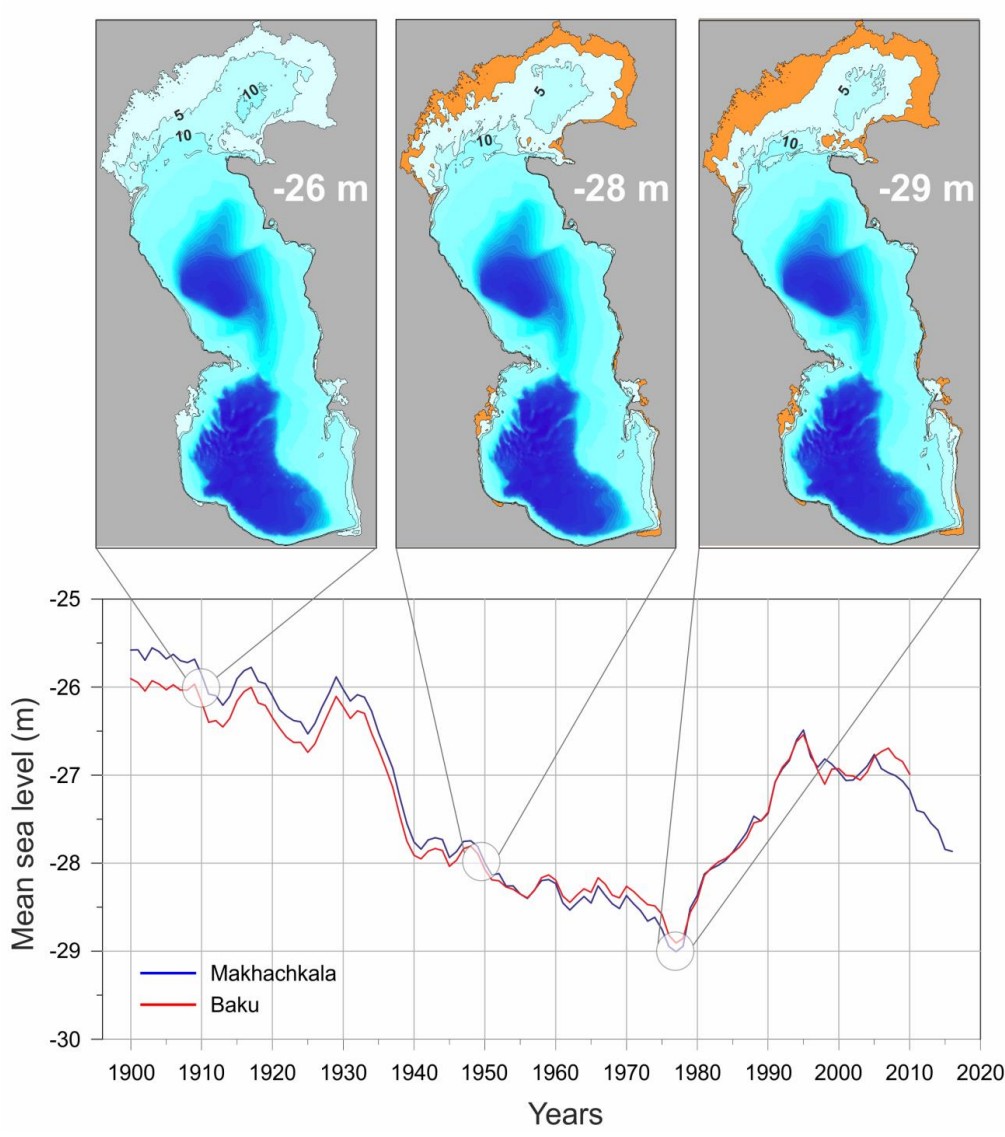

**Figure 5. Changes of the mean sea level (MSL) of the Caspian Sea at Makhachkala (blue line) and Baku (red line) and the bathymetry of the sea with the MSL -26, -28, and -29 m of the BHS. The orange areas are land territory created as a result of MSL changes.**

       Spatial structure of semidiurnal and diurnal is modified with the MSL changes of the Caspian Sea (Fig. 6). The amphidromic point shifts eastward by about 10 km with a decrease in the MSL from -25 to -29 m, it leads to general displacement of the area with amplitudes of 1.5–2 cm and also to the east. As a result, the $M_2$ amplitude decreases by 0.2–0.3 cm (up to 10–20% of amplitude)

10    on almost the entire eastern coast of the Middle Caspian. In the South Caspian, tidal amphidromy





also shifted to the east and amplitude increases at the western coast of the sea. An area of amplification of semidiurnal tides with amplitudes up to 6.5 cm is formed in the Mangyshlak Bay (North Caspian) with the MSL of -25 m. When the MSL drops to -28 m, amplitude in this bay decreases to 3.2 cm. An area of large amplitudes is again formed with a maximum of 5.5 cm in the Mangyshlak Bay (near the Tyuleniy Archipelago) with the MSL of 30 m.

The most interesting and complex modification of the tidal pattern occurs at the east coast of the sea. In the Türkmenbaşy Gulf, the amplitude decreases from 4.4 cm at the MSL of -25 m to 3.1 at the MSL of -29 m. The reverse picture is observed in the Turkmen Bay: the amplitude increases from 3.5 cm to 6.5 cm. The Turkmen Bay is a shallow semi-enclosed bay, with the Ogurja Ada Island situated on its western border. This island is a narrow sandy spit approximately 42 km long and 1–1.5 km wide. The island's height currently does not exceed 3–5 m (Badyukova, 2015). Thus, when the MSL of the Caspian Sea is -25 m, a significant part of the island is submerged. Results of our numerical experiments show that the presence of the island creates a western boundary in the Turkmen Bay. It leads to a change in frequency properties of the bay and as a consequence in an increase in the amplitude of semidiurnal tides.

More pronounced modification occurs in the diurnal tide pattern with the MSL changes. With the MSL of -25 m of the BHS, there is a more noticeable separation of amphidromy near the Absheron Peninsula into two separate systems: to the northeast and south of the peninsula. The amplitude of diurnal tide on the western coast of the South Caspian is 0.1–0.15 cm higher (up to 50% of amplitude) with the MSL of -29 m than for the MSL of -25 m. On the eastern coast of the South Caspian, the $K_1$ amplitude varies weakly with the MSL changes (by 10%). However, the $K_1$ phase lags are modified. It is caused by the influence of the Ogurja Ada Island with a low MSL.

Strong modifications of the diurnal tidal pattern with MSL changes in the mean sea level occur in the water area on the border of North and Middle Caspian. At the MSL of -25 m, the largest amplitude areas are located near the Tyuleniy Archipelago, up to 0.7–0.8 cm and in the Mangyshlak Bay, 1 cm. With the MSL decrease, large amplitudes area begins to expand to the west. At the MSL of -29 m, maximum amplitudes are already reached at the west coast of North Caspian (near the Tyuleny Island), up to 1.1 cm. These changes are apparently caused by strong modification of the bottom topography of the shallow North Caspian and as a result the frequency properties of this sea area.

25

30

Change in the spatial structure of the tidal range with the change in the MSL is similar to the $M_2$ amplitude pattern. The largest tidal range is in the Mangyshlak Bay, which is up to 22 cm (an MSL of -25 m). The tidal range in the Turkmen Bay is 13 cm, in the Türkmenbaşy Gulf is 15.5





cm. When the MSL decreases, the tidal range in Mangyshlak Bay decreases, and on the contrary, it increases in the Turkmen Bay. With the MSL of -29 m, the tidal range in the Turkmen Bay is 23 cm, whereas it is only 14 cm in the Mangyshlak Bay.



**Figure 6. Tidal maps for the amplitude of harmonic M₂ (a, b, c), K₁ (d, e, f), and tidal range (g, h, i) with different MSL of the Caspian Sea: -25 m (a, d, g), -27 m (b, e, h), -29 m (c, f, i).**





Changes in tidal characteristics can be very significant at individual sites. Figure 7 shows tidal vectors diagrams that display the $M_2$ model amplitude and phase lag for different sites at different MSL of the Caspian Sea. The amplitude and phase lag changes are relatively small at Makhachkala, Baku, and Bektash. The $M_2$ phase lag for Ogurja Ada changes by about 100°. The

5    $M_2$ amplitude changes twice: from 2.5 cm at MSL of -25 m to 5 cm at MSL of -30 m.

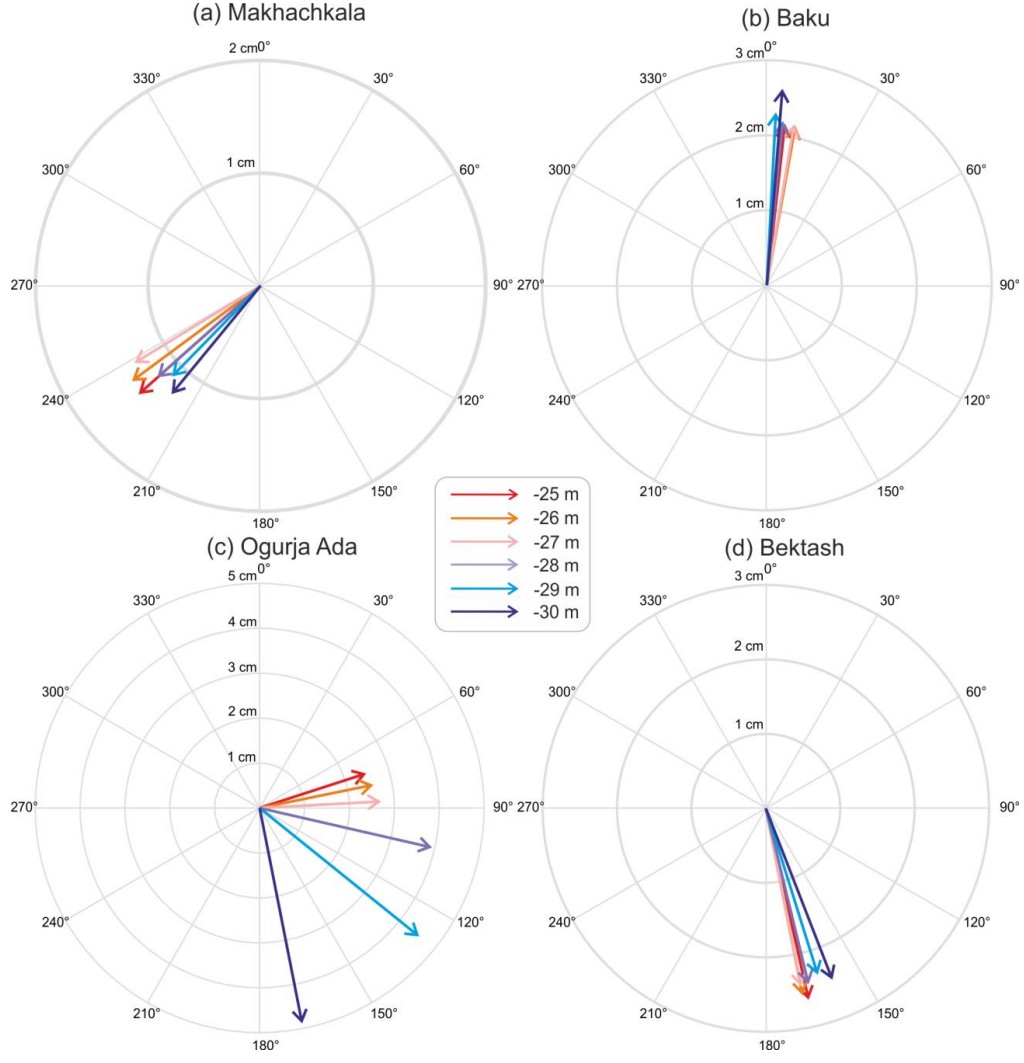

**Figure 7. The changes in the amplitude (cm) and phase lag (degrees) at four sites with different MSL based on numerical modelling results.**





**4 Discussion**

Results of numerical tidal modelling in this study are in good agreement with the results of harmonic analysis of tide gauge data of the Caspian Sea (Medvedev et al., 2017). Medvedev et al. (2017) demonstrated that the diurnal peak is absent in the sea level spectra for the western coast of the South Caspian (Baku, Svinoy Island), which is confirmed by the result of the numerical simulation the $K_1$ amplitude (Fig. 2a). Diurnal tides in the South Caspian are radiational and are formed under the influence of sea-breeze winds (Medvedev et al., 2017).

An unexpected result was obtained for the eastern part of South Caspian. With a high MSL (for example, -25 m) a significant part of the territory of the Ogurja Ada Island is below the water level. As a result, it makes it easier for tidal waves to penetrate the Turkmen Bay. With a low MSL (for example, -29 m), the area, or rather even the elongation of the island, increases significantly, and the island becomes an additional western border, reflecting tidal waves which penetrate the Turkmen Bay. According to (Badyukova, 2015), currently, the island's height (with the MSL of -27.5 m of the BHS) does not exceed 3–5 m. According to elevation data derived from the Shuttle Radar Topography Mission (SRTM, Farr et al., 2007), the island's maximum height is also 5-8 m. We used the GEBCO database to create our numerical grid for the model, with the island's maximum height being 2 m with MSL -28 m. Thus, in experiments with the MSL of -25 m, the island was completely below the water level. According to historical records in 1835, when the MSL of the Caspian Sea was -25.5 m, the central elevated part of the island was not flooded by the sea (the maximum height being about 3.5 m). According to (Badyukova, 2015), that island actually represents preserved fragments of a coastal delta plain which built on transgressive coastal bars and subsequently merged into one island. Comparison of the island's coordinates in 1850 with 2013 (Badyukova, 2015) shows that the island gradually has moved toward the land at the expense of redistribution of its deposits after erosion and changes of length and configuration. The greatest role in this process belongs to eolian processes. According to (Nikiforov, 1964), from one meter of the beach every hour, 5 kg of sand is carried inland with a wind speed of 4.9 km/s.

Numerical experiments were conducted with forcing produced by synthetic wind fields in order to assess changes in natural oscillations with a change in the MSL. Magnitude and direction of generated wind fields varied randomly every six hours. Spectral analysis of the simulated wind sea level variability showed that a decrease in the MSL leads to change in the period and Q-factor of natural oscillations of the Türkmenbaşy Gulf and the Turkmen Bay. When the MSL of the Caspian Sea in the Türkmenbaşy Gulf decreases, the Q-factor of seiches with a period of about 12 hours significantly decreases and at the MSL of -29 m, it does not exceed the spectral noise level (Fig. 8a). The Q-factor of natural oscillations of the bay with a period of about 7 hours increases.

In the Turkmen Bay decrease in the MSL from -26 m to -29 m, causes the spectral peak of the main seiche mode to migrate to the low-frequency band, thus the seiche period approaches the period of the harmonic $M_2$ (Fig. 8b). Apparently, this is due to elongation of the solid western boundary of the bay in the form of the Ogurja Ada Island. The closeness of the period of natural

oscillations to the tidal period (12.42 h) affects the structure of the tidal oscillations, thus the "sensitivity" of the tides to changes in the MSL is determined by the proximity or distance from the natural period.

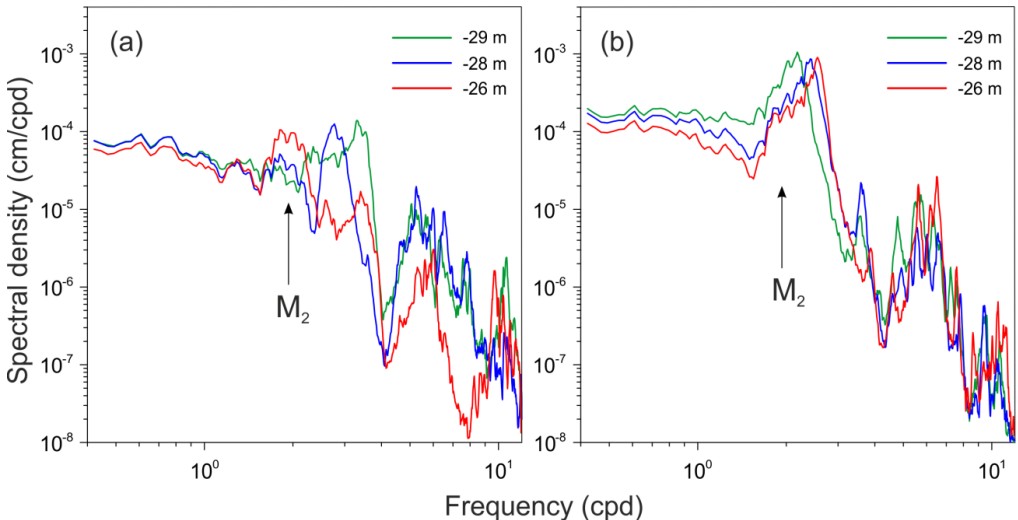

**Figure 8. Sea level spectra in (a) the Türkmenbaşy Gulf and (b) the Turkmen Bay at different**
**MSL of the Caspian Sea.**

## 5 Conclusions

In this study tidal dynamics of the Caspian Sea has been numerically investigated. The numerical simulation was forced by the direct action of the equilibrium tide. The main objective of the study was the mapping of tidal characteristics in the Caspian Sea. For the first time, it was

possible to construct detailed co-tidal maps of the tidal sea level and tidal current ellipses for the major harmonics using a numerical hydrodynamic model taking into account the data of long-term sea level observations. Results of numerical simulation indicate that maximum tidal amplitudes are located in the south-eastern part of the sea. We have shown that tidal currents can reach more than 20 cm/s in certain sea areas (for example, in straits), which is comparable to the magnitude

of permanent sea currents. It means that the role of tides in the water dynamic of isolated (non-tidal) seas seems to have been underestimated.



Numerical experiments indicate that the spatial features of tides are sensitive to the MSL changes. Modification of the tidal pattern is caused by changes in the bathymetry and geometry of the coastline of shallow areas of the sea including North Caspian, which results in the frequency response of the basin changes significantly. It is confirmed by the change in the natural oscillation (seiche) structure of the Caspian Sea.

In recent decades significant progress has been achieved in improvement of global barotropic tide models. This progress has been supported by available satellite altimetry. Stammer et al. (2014) detailed comparison of the main modern global barotropic tide models is presented. Some of these models include the Caspian Sea as well. But due to the lower MSL of the Caspian Sea relative to the global MSL, the results for the Caspian Sea in global tidal models have been seriously distorted.

We believe that our finding on tidal dynamics may be quite helpful in understanding the diurnal and semidiurnal variability in the sea level and currents in the Caspian Sea.

## 6 Data availability

Data and results in this article resulting from numerical simulations are available upon request from the corresponding author.

## 7 Author contribution

The concept of the study was jointly developed by IP and EK. IP did the numerical simulations, analysis, visualization and manuscript writing. EK prepared the numerical grids and participated in the analyses and the interpretation of the results. IF adapted the numerical Princeton Ocean Model (POM) to the Caspian Sea and participated in the verification stage. IP prepared the paper with contributions from EK and IF.

## 8 Competing interests

The authors declare that they have no conflict of interest.

## 9 Acknowledgements

This research was funded by the Russian Foundation for Basic Research, research project 18-05-01018 (numerical modelling of tidal elevations and currents) and the state assignment of IO RAS, theme 0149-2019-0005 (numerical experiments with different mean sea level).





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
