# Peer review of "Numerical modelling of the Caspian Sea tides"

_Ocean Science, 2019_

## Referee Comment (RC1) · Anonymous Referee #1 · 26 Aug 2019

GENERAL COMMENTS

The manuscript deals with an interesting, regionally relevant topic. The presented results and conclusions will certainly serve as a basis for future oceanographic, hydrological and geophysical investigations in the Caspian Sea environment. Given the particular sensitivity of the Caspian Sea and its mean water level to the changing climate, and in view of the current focus of the international research community on quantifying the variability of ocean tides over climatic time scales, especially the presented numerical experiments that reveal the dependence of both the tidal pattern and local seiches frequencies on the mean sea level is timely and highly relevant. Methods, results and conclusions are well presented. The paper is well structured; the amount and choice of figures, tables and equations is appropriate; the wording is generally adequate and

clear (some suggestions for rewording are given below). There are relatively few bibliographic references, but this might indicate in fact a shortage of previous work on this topic.

SPECIFIC COMMENTS

The scientific approach and the applied methods are valid and do, in my opinion, not require any correction.

Having myself a primarily observational background, I would have welcomed more information about the confrontation of the presented model with available sea-level data (tide gauges, altimetry). This is certainly a subjective preference, and I realize that this has been dealt in previous work (Medvedev et al. 2017, 2019: tide gauges) or may be the subject of upcoming publications (altimetry). Nevertheless, the reader could be provided with some additional interesting information, perhaps even in a qualitative way, adding a few sentences to the Introduction or Discussion. For example, what is the proportion of the relatively small astronomic tides in the total observable sea-level variation (i.e., how efficient is the presented model to predict real sea-level changes)?

The last paragraph of the Conclusions (p19,l6-11) is interesting and deserves more space. If the presented model indeed qualifies as an appropriate complement of global ocean tide models, this would be a mayor outcome of the paper and increase significantly its value. This question should be discussed in more detail. Purely empiric ocean tide models based on satellite altimetry should not be "distorted" by any assumption on the Caspian MSL. Here, again, rises the question about how well agree the presented model and altimetry, which could be offered as an outlook to future work or posed as an open research question. An invalid assumption on the Caspian MSL could indeed affect dynamical and assimilation models, but then it should be demonstrated which particular global ocean tide model assumes a wrong Caspian sea level and to which extent the tidal signal is distorted.

TECHNICAL CORRECTIONS
p1,l14: I understand that the splitting into two amphidromies occurs only in the diurnal case. If so, make this explicit: "For the diurnal constituents, the Absheron Peninsula splits this system into two separate amphidromies..." or so.

p1,l20-22: rephrase, e.g.: Numerical experiments with tidal simulation were made using different mean sea levels of the Caspian Sea (within a range of 5 m). The results indicate that the spatial features of the tides are strongly sensitive to changes of the mean sea level.

p1,l25: I prefer "one of the major drivers of ocean water motion" to "one of the major types".

p2,l1: "unique object for THE analysis"

p2,l5: "7.7 cm based on AN analysis of 30-day"

p2,l6: "performed A spectral analysis"

p2,l9: I prefer "Analyzing annual series..." to "Having analyzed annual series..."

p2,l14: "for different parts of the Caspian Sea" instead of "for different sea parts"

p2,l14: "... tide gauges. A maximum tidal range..."

p2,l16: "performed A high-resolution spectral analysis"

p2,l17: I prefer: "Southern (or Northern) Caspian" to "South (or North) Caspian" - throughout the text. Also "Central Caspian" to "Middle Caspian".

p2,l18: "of THE diurnal radiational constituent S1"

p2,l19: "than those of THE gravitational constituents"

p2,l22: "examination OF specific tidal features"

p2,l23: "in THE deep-water areas"

p2,l25: check reference format: "Caspian Sea (Medvedev et al. 2019)." or "Caspian

Sea in Medvedev et al. (2019)."

p3,l2: "we used A 2D version"

p3,l3: "in THE two-dimensional shallow water equations"

p3l12: check if this complies with the journal's reference format, or if there is another reference to this model to cite.

p3,l18: "THE energy dissipation of THE generated flows is caused by THE vertical turbulent viscosity. THE friction..."

p3,l22: "is THE flow velocity above..."

p4,l2: I prefer "to avoid a vanishing bottom drag in very deep waters"

p4,l4: "THE numerical simulations"

p4l6: "In section 3.1, a mean sea level (MSL) of the Caspian of -28 m with respect to the Baltic Height System (BHS, relative to the zero of the Kronstadt gauge) was adopted in the numerical modelling."

p4,l8: "from -25 m to -30 m with respect to the BHS. THE boundary conditions..." Consider replacing "** m of the BHS" by "** m with respect to the BHS" throughout the text.

p4,l11: reference format (see p2,l25); same in the following sentence

p5,Fig1: Kizlyar Bay is not indicated; also the tide gauges stations listed in Table 2 would be helpful to display. If necessary, the isobath annotation could be thinned out, or even omitted, if a color scale would be provided. In the caption, include a reference to the bathymetry model: "Figure 1: The bathymetry of the Caspian Sea according to ..."

p5,l8: "A numerical model with A MSL of -28 m with respect to the BHS..."

p6,l1: perhaps better "examine" instead of "consider"

p6,l2: "taking THE major constituents..."

p6,l3: "THE diurnal pattern includes..."

p6,l5: "have [or: feature] A counterclockwise rotation."

p6,l9: reference format (see p2,l25); I suggest rephrasing, e.g.: "Medvedev et al. (2019) showed that the results of numerical modelling are not really reliable in the Northern Caspian due to the very shallow depths in this area with about 20% of this part of the Caspian being less than 1 m deep (Baydin and Kosarev, 1986)."

p6,l12: "... have a spatial distribution similar to that of K1."

p6,l19: "The areas ... are: 1) the western part of the Southern Caspian..." or, better: "Maximum M2 amplitudes are found in 1) the western part of the Southern Caspian..."

p6,l24: "of THE major tidal constituents at selected cities [or: towns] around the Caspian Sea"

p7,Fig2: Add panel identifier "a)" and constituent identifier "K1" to the left panel.

p8,Tab2: After a quick glance at Fig. 2 it seems somewhat surprising that Fort Shevchenko features the largest tidal range. Perhaps this deserves a short comment.

p8,l7: "form factor AS determined by the AMPLITUDE ratio of the MAJOR diurnal and semidiurnal constituents"

p8,l11: "In general, A mixed mainly semidiurnal..."

p8,l12: "Only in THE western and eastern parts..."

p9,l3: "Based on the results of THE numerical modeling..."

p9,l6: I prefer "features a pattern similar to the M2 amplitude distribution"

p9,l9: I prefer here "are included" to "are presented"

p10,l3: "THE spatial structure..." I am used to the expression "semi-major axis" instead

of "major semi-axis" - revise throughout the text, same for semi-minor axis. "The largest M2 current velocities are observed [or: found] in:", or: "The areas of the largest M2 current velocities are:"

p10,l8: "up to 12.5 cm/s and 11.7 cm/s"

p10,l9: I prefer "depending on local topographic features."

p10,l11: "and THE tidal currents are nearly rectilinear."

p10,l12: "The spatial pattern of THE S2 tidal currents"

p10,l13: Perhaps "only the S2 semi-major axis is half of that of M2."

p10,l16: "repeats THE pattern of M2, too."

p10,l17: I prefer "by a factor of 1.8." to "by 1.8 times."

p11Fig4 caption: Semi-major axis velocity magnitudes (cm/s) for M2 tidal currents. Blue ellipses indicate clockwise circulation, red ellipses counterclockwise circulation.

p11,l4: I prefer: "3.4 Numerical experiments with VARYING MSL"

p11,l5: "THE interannual MSL variability..."

p11,l8: "and 20% of this area has a depth less than 1 m" - this has already been stated above, consider dropping this statement here.

p11,l9: I prefer "As a result, changes of the Caspian MSL by 2-3 m (as observed, e.g., between 1974 and 1994) lead to significant changes in the hydrodynamics of the Northern Caspian as well as in coastal waters of the Central and Southern Caspian."

p11,l11: "THE spatial characteristics of natural resonant oscillations in the basin (seiches)"

p11,l13: "experiments with tidal simulationS using..."

p11,l14: "This corresponds to the natural range of MSL changes..."

p11,l16: "THE results of these experiments allow to estimate [or: identify] the changes..."

p11,l18: "THE numerical results reveal that..."

p12,Fig5 caption: I prefer "Orange areas fall dry as a result of the assumed MSL changes."

p12,l6: "THE spatial structure of THE semidiurnal and diurnal TIDES is modified..."

p12,l8: "..from -25 m to -29 m, it leads to A general ..." "amplitudes of 1.5-2 cm and also to the east" is not clear - is there something missing?

p12,l10: I prefer "along almost the entire eastern shore" "In the Southern Caspian, THE tidal amphidromy also shifts to the east and THE amplitudeS increase along the western coast."

p13,l3: "...drops to -28 m, THE amplitude in this bay..."

p13,l5: "with a MSL of -30 m." (add the negative sign)

p13,l16: "A more pronounced modification occurs", or: "An even more pronounced modification", or: "More pronounced modifications occur"

p13,l19: "The amplitude of THE diurnal tide"

p13,l22: "THIS is caused ... AT low MSL."

p13,l23: rephrase, e.g.: "Strong modifications of the diurnal tidal pattern due to MSL changes occur along the transition between Northern and Central Caspian. At a MSL of -25 m the largest amplitudes are located..."

p13,l26: "With decreasing MSL, large amplitudes extent farther west."

p13,l27: "At a MSL of...", revise throughout the text.

p13,l28: I prefer "These changes are probably caused by a strong modification... and

,as a result, of the frequency [or: resonant] properties of this part of the sea [or: sub-basin]."

p13,l31: "THE change in the spatial structure"

p13,l32: I prefer "The maximum tidal range of 22 cm is found in the Mangyshlak Bay for a MSL of -25 m. At this MSL the tidal range in the Turkmen Bay amounts to 13 cm and in the Türkmenbaşy Gulf to 15.5 cm."

p16,l1: "The changes in tidal characteristics..."

p16,l2: "tidal vector diagrams... for different MSL of the Caspian Sea."

p16,l4: "However, the M2 phase lag ... 100° and the M2 amplitude doubles: from 2.5 cm at a MSL of ..."

p17,l1: "THE results of THE numerical tidal modelling... of A harmonic analysis..."

p17,l4: "... spectra FROM the western coast" (or: at)

p17,l6: "simulation of THE K1 amplitude"

p17,l8: "eastern part of THE Southern Caspian"

p17,l11: I prefer "..., the area and length of the island increase significantly, and the islands becomes an effective boundary to the west, reflecting THE tidal waves..."

p17,l13: reference format: Badyukova

p17,l17: I prefer "... , with a maximum elevation of the island of 2 m at a MSL of -28 m. Thus, in the experiments assuming a MSL of -25 m, the island was completely submerged."

p17,l22: "A comparison of the island's..."

p17,l23: I prefer "has gradually moved eastward and has changed its geometrical configuration due to the redistribution of deposits and erosion."

p17,l25: I prefer "The greatest contribution to this process originates from eolian redistribution." reference format: Nikiforov

p17,l28: "THE magnitude and direction of THE generated wind fields"

p17,l29: "A spectral analysis..."

p17,l31: rephrase, e.g.: When the MSL of the Caspian Sea decreases, the Q-factor of seiches with a period of about 12 hours significantly decreases in the Türkmenbaşy Gulf and at a MSL of -29 m it does not exceed any more the spectral noise level.

p18,l1: "In the Turkmen Bay a decrease in MSL from -26 m to -29 m causes the spectral peak of the main seiches mode to migrate towards lower frequencies..."

p18,l3: Consider dropping "Apparently,". "This is due to the progressive elongation of Ogurja Ada Island which represents the western boundary of the bay."

p18,l12: "In this study THE tidal dynamics of the Caspian Sea HAVE been numerically investigated."

p18,l17: "THE results of THE numerical simulation"

p18,l20: I prefer "hydrodynamics" to "water dynamic"; "... might have been underestimated so far."

p19,l1: "OUR numerical experiments indicate... sensitive to changes in the MSL. A modification..."

p19,l3: "including the Northern Caspian, which results in significant changes in the frequency response of the basin. This is also confirmed by..."

p19,l6: "in THE improvement of"

p19,l7: "Stammer et al. (2014) present a detailed comparison of the main modern global barotropic tide models."

p19,l12: I prefer "We believe that our findings on the tidal dynamics can help to better

understand the diurnal and semidiurnal variability in the sea level and currents in the Caspian Sea."

p19,l18 (Author contribution): It think that "IP" should be replaced by "IM" throughout section 7.

---

## Referee Comment (RC2) · Anonymous Referee #2 · 10 Sep 2019

SPECIFIC COMMENTS I generally find the paper and topic interesting but in many ways the paper is positioning itself between two chairs. One describing the result of the numerical modelling (not the numerical model) and the second the effect of lake level change. So given the title of the paper this was a bit of surprise to me.

So I would very much like to se the title reflecting this better like "ocean tides under changing lake level". In general I find the scientific approach and the applied methods valid though I have the same problem as the first reviewer that no information of the presented model is given as this is given in previous work. (Medvedev et al. 2017, 2019: tide gauges). I find the investigation of love numbers misplaced in this context as this is likely dealt with in the reference work, and I suggest this is substituted with more quantitative discussion on the quality of the model.

[Figure]

TECHNICAL Figure 1 is nice but identical to another publication by the leading author. As the evaluation of the ocean tide model in Table 2 is done for a number of cities surrounding the Caspian it would be much more appropriate if Figure 1 was changes to represent the location of these cities and I personally have no clue to where the cities are located. This would make reading easier.

Of interest I am very puzzled about the >21 cm tides in the TB described in Figure 3 because it does not relate very well to the amplitudes of the two major constituents in Figure 2 and the 4 major constituents in Table 2. the major semi+diurnal constituents explains a maximum of 7 cm or 1/3 of the tidal range in TB and the Maximum tidal range (R) for the 4 major explains less than 1/2 of the signal. Consequently there must be other major constituents not mentioned in this paper that is responsible and likely dominating?. Again the fact that I do not know the location of the cities in Table 2 makes it hard to determine the location of maximum amplitudes. The paper deserves an detailed explanation of this phaenomenas (is it astronomical constituents, overtides???).

The paper briefly mentions the form factor F in Table 2 and later in the paper gives one sentence about it. The form factor is detailed in previous publications by Medevedev, and I would leave it out of describe it much more detailed in this publication.

When discussing numerical experiments with different MSL more information on the accuracy of the bathymetry used must be provided. The discussion on Page 13 following Figure 6 is interesting but again I question on the Turkmen Bay.

Figure 4 could benefit from names on the regional features

Figure 5 6 and 7 should be reconsidered an redrawn for consistency. Figure 5 used 26 28 and 29 meters, Figure 6 25, 27 and 29 meters and Figure 7 25-30 meters. so they all are consistent.

Figure 5 also needs a bit of "regional" explanation for the reader. How can two cities 300 km apart. Exhibit sea level changes differing by 0.5 meters from 1900 until now.

[Figure]

Since 1980 the sea level curve matches but before it differs up to 0.5 meters?-

Figure 8 is interesting in attempting to explain the spectral density at different MSL regimes. I guess this is the key to the large tides in the Turkmen Bay, and the key to which constituents are responsible for the large tides. This deserved more attention and investigation and explanation in my oppinion.

In the discussion there is a bit of uncertainty to the discussion of the large tides in the Turkmen Bay. The height of the island is in the paper claimed to be 3-5 meters by the author and 5-8 meters from the SRTM. SRTM was meaured in the Early 2000's where sea level was -27.5 meters, so there is inconsistency here.
* * *

---

## Author Comment (AC1) · 22 Oct 2019

RESPONSE TO REVIEWER 1

Reviewer's comments are inserted in italics and blue, and responses in regular font.
Many thanks for these comments.

**GENERAL COMMENTS**

*The manuscript deals with an interesting, regionally relevant topic. The presented results and conclusions will certainly serve as a basis for future oceanographic, hydrological and geophysical investigations in the Caspian Sea environment. Given the particular sensitivity of the Caspian Sea and its mean water level to the changing climate, and in view of the current focus of the international research community on quantifying the variability of ocean tides over climatic time scales, especially the presented numerical experiments that reveal the dependence of both the tidal pattern and local seiches frequencies on the mean sea level is timely and highly relevant. Methods, results and conclusions are well presented. The paper is well structured; the amount and choice of figures, tables and equations is appropriate; the wording is generally adequate and clear (some suggestions for rewording are given below). There are relatively few bibliographic references, but this might indicate in fact a shortage of previous work on this topic.*

There are a few papers devoted to the problem of tides in the Caspian Sea. We tried to include in the review all the main papers on this topic.

**SPECIFIC COMMENTS**

*The scientific approach and the applied methods are valid and do, in my opinion, not require any correction.*

*Having myself a primarily observational background, I would have welcomed more information about the confrontation of the presented model with available sea-level data (tide gauges, altimetry). This is certainly a subjective preference, and I realize that this has been dealt in previous work (Medvedev et al. 2017, 2019: tide gauges) or may be the subject of upcoming publications (altimetry). Nevertheless, the reader could be provided with some additional interesting information, perhaps even in a qualitative way, adding a few sentences to the Introduction or Discussion. For example, what is the proportion of the relatively small astronomic tides in the total observable sea-level variation (i.e., how efficient is the presented model to predict real sea-level changes)?*

We have added in the current article a comparison of modeling results with the results of harmonic analysis according to observations on coastal tide gauges. In particular, we added one additional comparison figure for harmonics $M_2$ and $K_1$ and a short section with text.

We added a few paragraphs to the discussion reflecting the assessment of the contribution of tides to the variance of total sea level fluctuations with periods from 6 hours to 2 days. In the current research, we estimated the contribution of gravitational tides to the sea level variance based on the numerical modelling results. We made two numerical experiments: 1) with the tidal input; 2) with meteorological forcing produced by the fields of wind and air pressure variations over the Caspian Sea for 1979 from NCEP/CFSR reanalysis (Saha et al., 2010). We calculated the variance of tidal sea level variability (excluding long-period constituents) and the variance of the meteorological sea level variations in the first frequency band from 0.1 to 6 cpd and the second frequency band from 0.5 to 6 cpd. Then we estimated the relative contribution (in percent) of tides to the total sea level variance in the Caspian Sea.

The maximal contribution of tides to the total sea level variance has been located in the east part of the Middle Caspian: up to 29% for the first frequency band and up to 53% for the second frequency band. In the western part of the Southern Caspian and in Turkmen Bay the tidal contribution of total variance for the second frequency band from 0.5 to 6 cpd is up to 40%. The minimum contribution has been observed in the Northern Caspian, where strong storm surges occur; and near the Absheron Peninsula, where the amphidromic points of the diurnal and semidiurnal tides are located.

*The last paragraph of the Conclusions (p19,l6-11) is interesting and deserves more space. If the presented model indeed qualifies as an appropriate complement of global ocean tide models, this would be a mayor outcome of the paper and increase significantly its value. This question should be discussed in more detail. Purely empiric ocean tide models based on satellite altimetry should not be "distorted" by any assumption on the Caspian MSL. Here, again, rises the question about how well agree the presented model and altimetry, which could be offered as an outlook to future work or posed as an open research question. An invalid assumption on the Caspian MSL could indeed affect dynamical and assimilation models, but then it should be demonstrated which particular global ocean tide model assumes a wrong Caspian sea level and to which extent the tidal signal is distorted.*

We rewrote this paragraph a bit and tried to make it clearer. Most of the models presented by Stammer et al. (2014) don't include the Caspian Sea: FES14, EOT11a, TPXO9, GOT4.10, OSU12, DTU10, HAMTide. The TPXO9 included the Caspian Sea, but the MSL of the sea was 0 m with respect to the BHS. This invalid assumption shifted the coastline and significantly increased the sea area and as a result distorted the tide in this sea.

*TECHNICAL CORRECTIONS*

*p1,l14: I understand that the splitting into two amphidromies occurs only in the diurnal case. If so, make this explicit: "For the diurnal constituents, the Absheron Peninsula splits this system into two separate amphidromies..." or so.*

*p1,l20-22: rephrase, e.g.: Numerical experiments with tidal simulation were made using different mean sea levels of the Caspian Sea (within a range of 5 m). The results indicate that the spatial features of the tides are strongly sensitive to changes of the mean sea level.*

*p1,l25: I prefer "one of the major drivers of ocean water motion" to "one of the major types".*

*p2,l1: "unique object for THE analysis"*

*p2,l5: "7.7 cm based on AN analysis of 30-day"*

*p2,l6: "performed A spectral analysis"*

*p2,l9: I prefer "Analyzing annual series..." to "Having analyzed annual series..."*

*p2,l14: "for different parts of the Caspian Sea" instead of "for different sea parts"*

*p2,l14: "... tide gauges. A maximum tidal range..."*

*p2,l16: "performed A high-resolution spectral analysis"*

*p2,l17: I prefer: "Southern (or Northern) Caspian" to "South (or North) Caspian" - throughout the text. Also "Central Caspian" to "Middle Caspian".*

*p2,l18: "of THE diurnal radiational constituent S1"*

*p2,l19: "than those of THE gravitational constituents"*

*p2,l22: "examination OF specific tidal features"*

*p2,l23: "in THE deep-water areas"*

*p2,l25: check reference format: "Caspian Sea (Medvedev et al. 2019)." or "Caspian Sea in Medvedev et al. (2019)."*

*p3,l2: "we used A 2D version"*

*p3,l3: "in THE two-dimensional shallow water equations"*

*p3l12: check if this complies with the journal's reference format, or if there is another reference to this model to cite.*

*p3,l18: "THE energy dissipation of THE generated flows is caused by THE vertical turbulent viscosity. THE friction..."*

*p3,l22: "is THE flow velocity above..."*

*p4,l2: I prefer "to avoid a vanishing bottom drag in very deep waters"*

*p4,l4: "THE numerical simulations"*

*p4l6: "In section 3.1, a mean sea level (MSL) of the Caspian of -28 m with respect to the Baltic Height System (BHS, relative to the zero of the Kronstadt gauge) was adopted in the numerical modelling."*

*p4,l8: "from -25 m to -30 m with respect to the BHS. THE boundary conditions..." Consider replacing "** m of the BHS" by "** m with respect to the BHS" throughout the text.*

*p4,l11: reference format (see p2,l25); same in the following sentence*

*p5,Fig1: Kizlyar Bay is not indicated; also the tide gauges stations listed in Table 2 would be helpful to display. If necessary, the isobath annotation could be thinned out, or even omitted, if a color scale would be provided. In the caption, include a reference to the bathymetry model: "Figure 1: The bathymetry of the Caspian Sea according to..."*

*p5,l8: "A numerical model with A MSL of -28 m with respect to the BHS..."*

*p6,l1: perhaps better "examine" instead of "consider"*

*p6,l2: "taking THE major constituents..."*

*p6,l3: "THE diurnal pattern includes..."*

*p6,l5: "have [or: feature] A counterclockwise rotation."*

*p6,l9: reference format (see p2,l25); I suggest rephrasing, e.g.: "Medvedev et al. (2019) showed that the results of numerical modelling are not really reliable in the Northern Caspian due to the very shallow depths in this area with about 20% of this part of the Caspian being less than 1 m deep (Baydin and Kosarev, 1986)."*

*p6,l12: "... have a spatial distribution similar to that of K1."*

*p6,l19: "The areas ... are: 1) the western part of the Southern Caspian..." or, better: "Maximum M2 amplitudes are found in 1) the western part of the Southern Caspian..."*

*p6,l24: "of THE major tidal constituents at selected cities [or: towns] around the Caspian Sea"*

*p7,Fig2: Add panel identifier "a)" and constituent identifier "K1" to the left panel.*

*p8,Tab2: After a quick glance at Fig. 2 it seems somewhat surprising that Fort Shevchenko features the largest tidal range. Perhaps this deserves a short comment.*

*p8,l7: "form factor AS determined by the AMPLITUDE ratio of the MAJOR diurnal and semidiurnal constituents"*

*p8,l11: "In general, A mixed mainly semidiurnal..."*

*p8,l12: "Only in THE western and eastern parts..."*

*p9,l3: "Based on the results of THE numerical modeling..."*

*p9,l6: I prefer "features a pattern similar to the M2 amplitude distribution"*

*p9,l9: I prefer here "are included" to "are presented"*

p10,l3: "THE spatial structure..." I am used to the expression "semi-major axis" instead of "major semi-axis" - revise throughout the text, same for semi-minor axis. "The largest M2 current velocities are observed [or: found] in:", or: "The areas of the largest M2 current velocities are:"

p10,l8: "up to 12.5 cm/s and 11.7 cm/s"

p10,l9: I prefer "depending on local topographic features."

p10,l11: "and THE tidal currents are nearly rectilinear."

p10,l12: "The spatial pattern of THE S2 tidal currents"

p10,l13: Perhaps "only the S2 semi-major axis is half of that of M2."

p10,l16: "repeats THE pattern of M2, too."

p10,l17: I prefer "by a factor of 1.8." to "by 1.8 times."

p11Fig4 caption: Semi-major axis velocity magnitudes (cm/s) for M2 tidal currents. Blue ellipses indicate clockwise circulation, red ellipses counterclockwise circulation.

p11,l4: I prefer: "3.4 Numerical experiments with VARYING MSL"

p11,l5: "THE interannual MSL variability..."

p11,l8: "and 20% of this area has a depth less than 1 m" - this has already been stated above, consider dropping this statement here.

p11,l9: I prefer "As a result, changes of the Caspian MSL by 2-3 m (as observed, e.g., between 1974 and 1994) lead to significant changes in the hydrodynamics of the Northern Caspian as well as in coastal waters of the Central and Southern Caspian."

p11,l11: "THE spatial characteristics of natural resonant oscillations in the basin (seiches)"

p11,l13: "experiments with tidal simulationS using..."

p11,l14: "This corresponds to the natural range of MSL changes..."

p11,l16: "THE results of these experiments allow to estimate [or: identify] the changes..."

p11,l18: "THE numerical results reveal that..."

p12,Fig5 caption: I prefer "Orange areas fall dry as a result of the assumed MSL changes."

p12,l6: "THE spatial structure of THE semidiurnal and diurnal TIDES is modified..."

p12,l8: "..from -25 m to -29 m, it leads to A general ..." "amplitudes of 1.5-2 cm and also to the east" is not clear - is there something missing?

p12,l10: I prefer "along almost the entire eastern shore" "In the Southern Caspian, THE tidal amphidromy also shifts to the east and THE amplitudeS increase along the western coast."

p13,l3: "...drops to -28 m, THE amplitude in this bay..."

p13,l5: "with a MSL of -30 m." (add the negative sign)

p13,l16: "A more pronounced modification occurs", or: "An even more pronounced modification", or: "More pronounced modifications occur"

*p13,l19: "The amplitude of THE diurnal tide"*

*p13,l22: "THIS is caused ... AT low MSL."*

*p13,l23: rephrase, e.g.: "Strong modifications of the diurnal tidal pattern due to MSL changes occur along the transition between Northern and Central Caspian. At a MSL of -25 m the largest amplitudes are located..."*

*p13,l26: "With decreasing MSL, large amplitudes extent farther west."*

*p13,l27: "At a MSL of...", revise throughout the text.*

*p13,l28: I prefer "These changes are probably caused by a strong modification... and, as a result, of the frequency [or: resonant] properties of this part of the sea [or: subbasin]."*

*p13,l31: "THE change in the spatial structure"*

*p13,l32: I prefer "The maximum tidal range of 22 cm is found in the Mangyshlak Bay for a MSL of -25 m. At this MSL the tidal range in the Turkmen Bay amounts to 13 cm and in the Türkmenba¸sy Gulf to 15.5 cm."*

*p16,l1: "The changes in tidal characteristics..."*

*p16,l2: "tidal vector diagrams... for different MSL of the Caspian Sea."*

*p16,l4: "However, the M2 phase lag ... 100_ and the M2 amplitude doubles: from 2.5 cm at a MSL of ..."*

*p17,l1: "THE results of THE numerical tidal modelling... of A harmonic analysis..."*

*p17,l4: "... spectra FROM the western coast" (or: at)*

*p17,l6: "simulation of THE K1 amplitude"*

*p17,l8: "eastern part of THE Southern Caspian"*

*p17,l11: I prefer "..., the area and length of the island increase significantly, and the islands becomes an effective boundary to the west, reflecting THE tidal waves..."*

*p17,l13: reference format: Badyukova*

*p17,l17: I prefer "... , with a maximum elevation of the island of 2 m at a MSL of -28 m. Thus, in the experiments assuming a MSL of -25 m, the island was completely submerged."*

*p17,l22: "A comparison of the island's..."*

*p17,l23: I prefer "has gradually moved eastward and has changed its geometrical configuration due to the redistribution of deposits and erosion."*

*p17,l25: I prefer "The greatest contribution to this process originates from eolian redistribution." reference format: Nikiforov*

*p17,l28: "THE magnitude and direction of THE generated wind fields"*

*p17,l29: "A spectral analysis..."*

*p17,l31: rephrase, e.g.: When the MSL of the Caspian Sea decreases, the Q-factor of seiches with a period of about 12 hours significantly decreases in the Türkmenba¸sy Gulf and at a MSL of -29 m it does not exceed any more the spectral noise level.*

*p18,l1: "In the Turkmen Bay a decrease in MSL from -26 m to -29 m causes the spectral peak of the main seiches mode to migrate towards lower frequencies..."*

*p18,l3: Consider dropping "Apparently,". "This is due to the progressive elongation of Ogurja Ada Island which represents the western boundary of the bay."*

*p18,l12: "In this study THE tidal dynamics of the Caspian Sea HAVE been numerically investigated."*

*p18,l17: "THE results of THE numerical simulation"*

*p18,l20: I prefer "hydrodynamics" to "water dynamic"; "... might have been underestimated so far."*

*p19,l1: "OUR numerical experiments indicate... sensitive to changes in the MSL. A modification..."*

*p19,l3: "including the Northern Caspian, which results in significant changes in the frequency response of the basin. This is also confirmed by..."*

*p19,l6: "in THE improvement of"*

*p19,l7: "Stammer et al. (2014) present a detailed comparison of the main modern global barotropic tide models."*

*p19,l12: I prefer "We believe that our findings on the tidal dynamics can help to better understand the diurnal and semidiurnal variability in the sea level and currents in the Caspian Sea."*

*p19,l18 (Author contribution): It think that "IP" should be replaced by "IM" throughout section 7.*

We agree with all comments in technical corrections section and clarified these sentences.

Some comments:

The reviewer correctly noted that Fort Shevchenko features the largest tidal range among other cities in Table 2. The maximum tidal range has been observed in the Turkmen Bay (21 cm), but there are no major cities in this area. We added short comment about it in text of manuscript.

---

## Author Comment (AC2) · 22 Oct 2019

RESPONSE TO REVIEWER 2

Reviewer's comments are inserted in italics and blue, and responses in regular font.

Many thanks for these comments.

**SPECIFIC COMMENTS**

*I generally find the paper and topic interesting but in many ways the paper is positioning itself between two chairs. One describing the result of the numerical modelling (not the numerical model) and the second the effect of lake level change. So given the title of the paper this was a bit of surprise to me.*

*So I would very much like to se the title reflecting this better like "ocean tides under changing lake level". In general I find the scientific approach and the applied methods valid though I have the same problem as the first reviewer that no information of the presented model is given as this is given in previous work. (Medvedev et al. 2017, 2019: tide gauges). I find the investigation of love numbers misplaced in this context as this is likely dealt with in the reference work, and I suggest this is substituted with more quantitative discussion on the quality of the model.*

Our paper describes the result of the numerical modelling. We added in new version of manuscript information about the confrontation of the presented model with tide gauge data.

We believe that the main results of presented paper tidal charts for amplitudes and phase lags of the major tidal harmonics, form factor, tidal range and velocity of tidal currents. The numerical results with the changes of the mean sea level are secondary. Therefore, we believe that the current title of the article reflects well the results presented in it. We didn't do the investigation the Love numbers, but simply describe the model parameters.

**TECHNICAL**

*Figure 1 is nice but identical to another publication by the leading author. As the evaluation of the ocean tide model in Table 2 is done for a number of cities surrounding the Caspian it would be much more appropriate if Figure 1 was changes to represent the location of these cities and I personally have no clue to where the cities are located. This would make reading easier.*

We corrected Figure 1 and added some names of bays and cities from Table 2.

*Of interest I am very puzzled about the >21 cm tides in the TB described in Figure 3 because it does not relate very well to the amplitudes of the two major constituents in Figure 2 and the 4 major constituents in Table 2. the major semi-diurnal constituents explains a maximum of 7 cm or 1/3 of the tidal range in TB and the Maximum tidal range (R) for the 4 major explains less than 1/2 of the signal. Consequently there must be other major constituents not mentioned in this paper that is responsible and likely dominating?. Again the fact that I do not know the location of the cities in Table 2 makes it hard to determine the location of maximum amplitudes. The paper deserves an detailed explanation of this phenomena (is it astronomical constituents, overtides???).*

We think that there is no surprise in the estimates of the tidal range. The tidal range was calculated as the maximum range of tidal sea level oscillations during one lunar day (~25 hours). This is approximately equal to twice the sum of the four major constituents (M2, S2, K1, and O1). For example in Table 2 for Fort Shevchenko we have $H$(M2)= 2.47 cm, $H$(S2)= 0.92 cm, $H$(K1)= 0.56 cm, $H$(O1)= 0.30 cm. The twice sum of these constituents is 8.5 cm, that is relate

well to the tidal range in Table 2 for this city (8.9 cm). In Turkmen Bay (see Fig. 3 in new version of paper) we have $H$(M2)= 6 cm, $H$(S2)= 2.6 cm, $H$(K1)= 0.73 cm, $H$(O1)= 0.47 cm. The twice sum of these constituents is 19.6 cm, that is relate well to the tidal range = 21 cm in Fig. 4. The differences in the magnitude of the tidal range in paper and presented twice sums are caused by the contribution of semidiurnal constituents N2 and K2 with amplitudes of about 0.5-1 cm.

*The paper briefly mentions the form factor F in Table 2 and later in the paper gives one sentence about it. The form factor is detailed in previous publications by Medevedev, and I would leave it out of describe it much more detailed in this publication.*

In current paper, we show for the first time a map of form factor for the Caspian Sea. Consequently, we will keep the short description of it.

*When discussing numerical experiments with different MSL more information on the accuracy of the bathymetry used must be provided. The discussion on Page 13 following Figure 6 is interesting but again I question on the Turkmen Bay.*

*Figure 4 could benefit from names on the regional features*

*Figure 5 6 and 7 should be reconsidered an redrawn for consistency.*

*Figure 5 used 26 28 and 29 meters, Figure 6 25, 27 and 29 meters and Figure 7 25-30 meters. so they all are consistent. Figure 5 also needs a bit of "regional" explanation for the reader. How can two cities 300 km apart. Exhibit sea level changes differing by 0.5 meters from 1900 until now.*

*Since 1980 the sea level curve matches but before it differs up to 0.5 meters?-*

We added some information on the accuracy of the bathymetry.

We added names on the regional geographical features.

We redrawn Figure 5. We done new bathymetry maps for MSL = -25, -27, -29 m. The difference in the mean sea level of 0.3 m in Baku and Makhachkala caused us questions too. We wanted to show the mean sea level of the whole Caspian Sea, but since it is different depending on the station, we decided to show two stations in the figure. We checked several sources of data on the Caspian level (http://www.caspcom.com/ и http://caspi.ru/). In both databases the data shown in Fig. 5 differences in the average level at 0.3 m in the first half of the 20th century. We believe that this feature in interannual sea level variability is caused by the local conditions of these stations (for example, tectonic movements), which led to a relative change in the absolute height of the tidal pole. But since these questions are not the purpose of this study, we decided to show in this figure only one station (Makhachkala).

*Figure 8 is interesting in attempting to explain the spectral density at different MSL regimes. I guess this is the key to the large tides in the Turkmen Bay, and the key to which constituents are responsible for the large tides. This deserved more attention and investigation and explanation in my oppinion.*

We have added a few more words to this section.

*In the discussion there is a bit of uncertainty to the discussion of the large tides in the Turkmen Bay. The height of the island is in the paper claimed to be 3-5 meters by the author and 5-8 meters from the SRTM. SRTM was measured in the Early 2000's where sea level was -27.5 meters, so there is inconsistency here.*

We agree with the reviewer that there is inconsistency in the height of the Ogurja Ada Island. The SRTM was in February 2000 where sea level was -27 m. When we took the SRTM data we expected to see the height of the island about 1-3 m, but it turned out to be higher. Because of this difference between the results of Badyukova, 2015, GEBCO database and SRTM data we put this paragraph in the discussion. We have not yet found more reliable information about the height of the island and now we can't say who is right. We will try to study this in more detail in the subject of upcoming future researches.

Many thanks.

---

## Editor Comment (EC1) · Philip Woodworth (Editor) · 23 Oct 2019

For the record, I should say that I largely agree with the two reviews.

R1 seems to contradict itself slightly in saying the paper is 'well presented' but it then provides a long (but very useful) set of technical corrections. I did not think it was well presented, see next paragraph. Also the reviewer did not like the way the Conclusions were presented.

I had many small corrections also myself which I sent privately to the author, so I am hoping that any new version will be much improved technically.

Otherwise, I also had a problem with the way the Conclusions were written. I think what the author was trying to say is that he computed the spectra of the Caspian (in

the 2017 paper), and that variation of MSL changes the spectra such that the main spectral peak moves closer or further away from the M2 frequency. I think that part has to be clearer.

Also I did not understand the last sentence of the penultimate paragraph of the Conclusions about global models being distorted. Maybe this is a language problem. I would reword this to read simply something like: Some of these models include the Caspian Sea but some do not. We believe that our findings ..

R2 had more fundamental remarks, even as to the title of the paper, and a general problem being how this paper overlaps with a previous one on the Caspian by the author. He also had many problems with the figures.

I note that a new version of the paper has today (23 October 2019) been made available. I will probably pass it by the reviewers again and of course read it myself with great interest.

---

## Author Response (AR2)

RESPONSE TO EDITOR

We are very grateful to the editor and reviewers for useful comments on the manuscript. We agree with all comments in your and Reviewer's technical corrections and clarified these sentences. All authors of this paper once again carefully read the text and made some changes. After which the English native speaker carefully read the article and corrected some features of the language. We hope that the quality of English in our article has grown.

In the new version of the manuscript, we removed the sentence about TPX08. The Caspian Sea is masked in the last version of this model (TPXO9). We decided to write that there is no Caspian Sea in global models, so as not to confuse the potential readers. (see Conclusions).

Also, we corrected Figures 7, 9, 10 in accordance with the Editor's comments. We removed the second frequency band (0.5-6 cpd) in the analysis of the tidal contribution to the total sea level variance. We redrawn the Figure 10 for MSL = -25 m, -27 m, -29 m.

In the process of revision, we decided to change the title of the article:

"Numerical modelling of the Caspian Sea tides"

Many thanks.